# Back to full interseismic plate locking decades after the giant 1960 Chile earthquake

Daniel Melnick [1,2], Shaoyang Li [3,4], Marcos Moreno [2,3,5], Marco Cisternas[2,6], Julius Jara-Muñoz[7], Robert Wesson[8], Alan Nelson[8], Juan Carlos Báez[9] & Zhiguo Deng[3]

Great megathrust earthquakes arise from the sudden release of energy accumulated during centuries of interseismic plate convergence. The moment deficit (energy available for future earthquakes) is commonly inferred by integrating the rate of interseismic plate locking over the time since the previous great earthquake. But accurate integration requires knowledge of how interseismic plate locking changes decades after earthquakes, measurements not available for most great earthquakes. Here we reconstruct the post-earthquake history of plate locking at Guafo Island, above the seismogenic zone of the giant 1960 ($M_w = 9.5$) Chile earthquake, through forward modeling of land-level changes inferred from aerial imagery (since 1974) and measured by GPS (since 1994). We find that interseismic locking increased to ~70% in the decade following the 1960 earthquake and then gradually to 100% by 2005. Our findings illustrate the transient evolution of plate locking in Chile, and suggest a similarly complex evolution elsewhere, with implications for the time- and magnitude-dependent probability of future events.

[1] Instituto de Ciencias de la Tierra, TAQUACh, Universidad Austral de Chile, Valdivia 5111430, Chile. [2] Millennium Nucleus The Seismic Cycle Along Subduction Zones, Valdivia, Concepción, Valparaíso. 5111430, Chile. [3] GFZ Helmholtz Zentrum Potsdam, Potsdam 14473, Germany. [4] Department of Earth and Environmental Sciences, University of Iowa, Iowa 52242 IA, USA. [5] Departamento de Geofísica, Universidad de Concepción, Concepción 160-C, Chile. [6] Escuela de Ciencias del Mar, Universidad Católica de Valparaíso, Valparaíso 1020, Chile. [7] Institut für Erd- und Umweltwissenschaften, Universität Potsdam, Potsdam 14476, Germany. [8] Geologic Hazards Science Center, U.S. Geological Survey, Denver 80225 CO, USA. [9] Centro Sismológico Nacional, Universidad de Chile, Facultad de Ciencias Físicas y Matemáticas, Santiago 8370448, Chile. Correspondence and requests for materials should be addressed to D.M. (email: daniel.melnick@uach.cl)

Great earthquakes are preceded by a period of seismic quiescence, when energy accumulates over decades to centuries during the interseismic period of the seismic cycle. Deducing the history of interseismic energy accumulation helps us understand seismogenic processes and so improve time-dependent hazard models used to estimate the probability and magnitude of future earthquakes. In subduction zones, where Earth's greatest earthquakes occur, the degree of locking between the incoming oceanic plate and overriding continent is commonly used to infer the seismic potential of regions of the continental margin[1–4]. Space geodesy has shown that the regional distribution of interseismic plate locking in the decade before a great earthquake is–to a certain degree–correlated with the distribution of coseismic slip during the earthquake[4–7]. The resemblance of these inter- and coseismic distributions emphasizes their importance for hazard assessment and for understanding the evolution of interseismic plate locking and its controlling processes.

Recent geodetic and seismologic studies suggest that the degree of interseismic plate locking may change several years to a few months before a great earthquake[8–11]. Because the coseismic slip distribution usually resembles the spatial pattern of interseismic locking, recognizing these changes in plate locking—and thus the beginning of a pre-seismic phase of an earthquake cycle—might significantly improve estimates of the time-dependent probability of the occurrence of a future great earthquake as well as its magnitude. Detecting such a pre-seismic phase requires long time series to isolate a background secular signal, which have been difficult to obtain for most subduction zones.

Modern rates of interseismic plate locking are commonly estimated by inverting surface velocities deduced from Global Positioning System (GPS) measurements[1,5,12] under the back-slip assumption[13]. Major measurement challenges are posed by the offshore location of the seismogenic zone—the portion of the subduction megathrust that slips during earthquakes and the locus of interseismic plate locking–where monitoring requires sophisticated acoustic technology[14,15]. Given the century or longer recurrence of great subduction earthquakes[16], estimating interseismic plate locking over several decades is an additional challenge, which relies mostly on rates of relative land- and sea-level change inferred from scattered leveling lines and tide-gauge stations[17]. But rare, isolated islands off the mainland coast directly above the seismogenic zone may record high-amplitude (meter-scale) interseismic deformation over decade-to-century time spans[18,19].

Recent studies suggest interseismic plate locking rates vary over time. Paleogeodetic records inferred from corals at islands above the Sumatran seismogenic zone show punctuated accelerations in relative sea level (RSL) that may reflect changes in interseismic plate locking over centuries[3,20]. Similar inferences based on RSL changes estimated from resurveyed nautical charts in south-central Chile suggest interseismic plate locking may not be re-established until decades after a great earthquake[19]. In contrast, GPS measurements suggest full interseismic plate locking was restored only 2 and 4 years after $M_w$ 7.2 and 8.0 earthquakes in Japan[21] and Mexico[22], respectively. On such short timescales, post-seismic processes, such as afterslip and mantle relaxation, may obscure the temporal restoration of interseismic plate locking[23]. We know little about the evolution of interseismic plate locking before great to giant earthquakes ($M_w$ 8.5–9.5), and its possible controlling factors, mostly due to the lack of decadal-scale geodetic data from sites directly above the seismogenic zone.

The Chilean subduction zone, where the oceanic Nazca Plate is sliding beneath South America at ~66 mm year$^{-1}$, has produced many great earthquakes ($M_w > 8$) during the past ~500 years of historical records[24]. The greatest of these, the giant 1960 Valdivia earthquake of $M_w$ 9.5, ruptured ~1000 km of the south-central

Chilean margin[25]. Coseismic uplift of the outer shelf area and on scattered islands, sudden subsidence of the mainland coast[25], deposits left by tsunamis, and shaking-induced beds of mud in inland lakes[26–29] suggest such giant events have recurred every ~285 years during the late Holocene. The 1960 rupture zone has been seismically quiet and characterized by a patchy spatial pattern of heterogeneous interseismic plate locking, with strongly locked areas segmented by creeping parts, as estimated using GPS velocities collected mostly during 2002–2010[12]. The 2016 $M_w$ 7.6 earthquake, which broke a small patch of the deep seismogenic locked zone beneath Chiloé Island's southern coast (Fig. 1b), announced the seismic reawakening of the 1960 rupture zone[30,31]. Regional interseismic plate locking estimated from horizontal components of 17 campaign and 6 continuous GPS stations suggest a high degree of locking below southern Chiloé and Guafo islands over a much larger region than the 2016 rupture zone[12,30].

Guafo Island lies above the upper part of the seismogenic zone as inferred from thermal modeling[32] and inversion of regional GPS velocities[12]. Because of its trenchward location, the amplitude of seismic and interseismic deformation on Guafo is greater than at more inland sites and, therefore, the island is more sensitive to changes in the degree of interseismic plate locking. Such high-amplitude deformation was directly observed during the 1960 earthquake by Navy officers at the Guafo lighthouse: coseismic uplift of 3.6–4.0 m[25,33]. At the same time, most of the mainland coast adjacent to the rupture zone, including Chiloé and the Guaitecas Archipelago (where continuous GPS stations CSTO and MELK are located, Fig. 1), subsided as much as 2 m[25].

Here we reconstruct the history of interseismic plate locking during the past 55 years at Guafo Island (Fig. 2), located above the south-central Chile seismogenic zone and at only 60 km from the trench (Fig. 1), using land-level changes estimated from aerial imagery, campaign and continuous GPS, and numerical modeling. Our results show how interseismic plate locking rates evolve, which has implications on the development of time-variable seismic hazard models.

## Results

**RSL changes since the 1960 earthquake.** In order to reconstruct the history of RSL change at Guafo, we analyzed the morphology of the coastline before the 1960 earthquake and during the following decades using nautical charts, aerial imagery, and satellite imagery. The pre-earthquake coastline was characterized by scattered sea stacks (rock pinnacles) along the rocky coast near the lighthouse, a narrow beach ridge partially damming the Caleta Rica river mouth, and small stream valleys flooded by the sea (Supplementary Figs. 2-5). Visual comparisons with post-earthquake data highlight the severity of coastline changes caused by coseismic uplift, and suggest that the coast was drowning through gradual RSL rise prior to the earthquake. The sudden drop in RSL caused by coseismic uplift in 1960 left an inlet dry, as deduced from comparisons among a 1929 nautical chart, 1944 aerial imagery, and post-earthquake imagery (Supplementary Fig. 3). The scattered sea stacks on the 1929 chart are today linked by a narrow emerged platform cut on resistant conglomerate bedrock (Supplementary Fig. 3). In areas of softer Tertiary sedimentary bedrock around the island, emerged abrasion platforms as much as 200 m wide have low relief with gentle slopes (Supplementary Figs. 2, 4, 5).

Presently, the bedrock platform at Caleta Rica is covered by scattered remnants of a peaty soil (i.e., the AO horizon) in the intertidal zone and by narrow fringes of beach sand along the high-tide line (Supplementary Figs. 2d, 3d). We found a plastic package labeled Fortesan (a soya-based nutritional complement

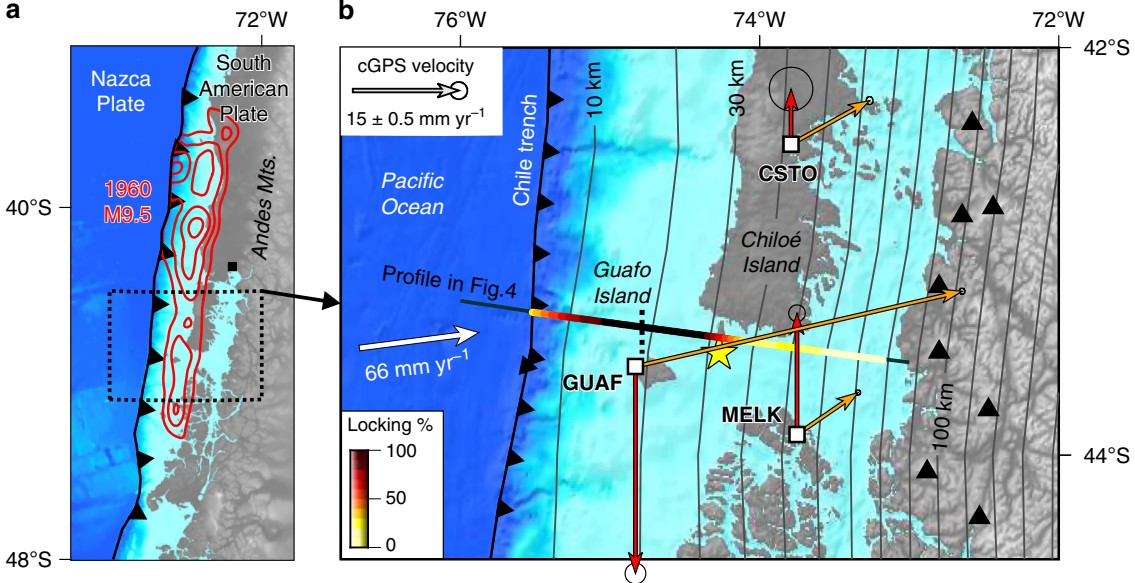

**Fig. 1** Tectonic setting of the 1960 Chile earthquake. **a** Megathrust slip contours[60] (red lines, 10-m interval). Black square shows location of Puerto Montt. **b** Arrows show vertical and horizontal velocities from continuous Global Position System (cGPS) stations with respect to a stable South American reference frame (Methods; time series in Supplementary Fig. 10). Yellow star shows epicenter of the 2016 Chiloé earthquake[30]. Triangles denote active volcanoes. White arrow shows plate convergence vector estimated from space geodesy. Interseismic plate locking estimated from inverse modeling of cGPS velocities expressed as percentage of plate convergence rate (Methods). Dark gray contours show 10-km depth intervals to the top of the subducting slab[36]. Note that the downdip limit of interseismic plate locking occurs at depths of ~30–35 km. Bathymetry and topography from srtm15plus data[64] (available from topex.ucsd.edu)

distributed only during the first years of Chile's military government in the mid 1970s) in the soil (Supplementary Fig. 6). This artifact dates the lower part of the soil horizon to shortly after emergence of the platform in 1960. Rapid soil development on intertidal areas that emerged in 1960 was likely boosted by the high mean annual precipitation (~3000 mm) and abundant supply of nutrients from the steep slopes of the adjacent rainforest (Fig. 3d). This inference is supported by observations made at Guamblin Island (~100 km south of Guafo) by George Plafker in 1968 that documented the incipient development of new soil on a former marine surface[25].

At Caleta Rica (Fig. 2), 1974 aerial imagery shows that the emerged bedrock platform had been rapidly colonized by a wet meadow/fen, probably with rushes, grasses, and bushes (Supplementary Fig. 7). By 2008, however, during our first visit to the island, the lower part of the soil AO horizon was being eroded by tides, and locally, dead growth-position roots and stumps of young trees and bushes killed by tides were still attached to the platform (Figs. 2d and 3b–d). Immediately above high tide, we found a plant community of young, gray, defoliated trees, including *Luma apiliculata* and *Drimys winteri*, together with healthy salt-tolerant rushes (*Juncus*) and cordgrass (*Spartina*). Landward and above the defoliated trees, similar communities are healthy suggesting the edge of the lowland fen is slowly dying as a result of saltwater inundation caused by RSL rise (Fig. 3a).

Thus, stratigraphic, geomorphic, and ecologic evidence, combined with imagery comparisons, show that RSL on Guafo fell suddenly in 1960 as a result of coseismic uplift, and was followed by RSL rise in subsequent decades. The post-1960 soil on the platform buries sessile organisms that lived attached to the bedrock before the earthquake. Today, well preserved remains of lower intertidal flora (*Lithothamnum*, coralline algae) and fauna (*Petricola dactylus* and *Petricola patagonica*, borer bivalves) have been exhumed by erosion of the soil (inset in Fig. 3d). The fact that the post-1960 soil developed above the bivalves shows that

the lower intertidal platform where they lived was raised above high tide after the earthquake; the current tidal erosion of the soil requires subsequent slow RSL rise.

**Land-level change rates after the 1960 earthquake**. To quantify the rapid post-1960 subsidence on Guafo, we combined the results of our historical imagery analysis with numerical models tuned by campaign and continuous GPS measurements (Figs. 4 and 5). First, we estimated the rate of post-1960 RSL rise by mapping successive positions of the shoreline (i.e., limit between dark bedrock abrasion platform and light sandy beach) on aerial imagery (Methods, Fig. 2c, Supplementary Fig. 7). We estimated the vertical displacement rate from the horizontal inland shift of the shoreline using local slopes (Supplementary Fig. 8) along 400 topographic profiles oriented normal to the shoreline (Fig. 2b); in this way, we obtained distributions of RSL rise rates from each image pair (Figs. 2b, 5a and Supplementary Fig. 9). These RSL rise rates were then converted to land-level change rates by subtracting a mean absolute sea-level change rate determined from satellite altimetry (Fig. 5b). We validate our rates from aerial imagery by comparing them with land-level change rates deduced from the campaign GPS benchmark GAFO (installed in 1994 and resurveyed in 2009) and from the continuous GPS station GUAF that we installed in 2009 (location in Fig. 2a; time series in Supplementary Fig. 10). These combined land-level change rates suggest Guafo has been subsiding continuously at an increasing rate of ~8 to 16 mm year$^{-1}$ since at least the l970s, with a mean acceleration rate of $0.14 \pm 0.08$ mm year$^{-2}$ (Fig. 5c).

**History of interseismic locking rate from back-slip modeling**. From our history of land-level change we infer the evolution of interseismic plate locking under certain assumptions regarding the kinematics of the interplate seismogenic zone. In order to isolate the component of deformation caused by interseismic plate locking we first subtract the post-seismic viscoelastic model

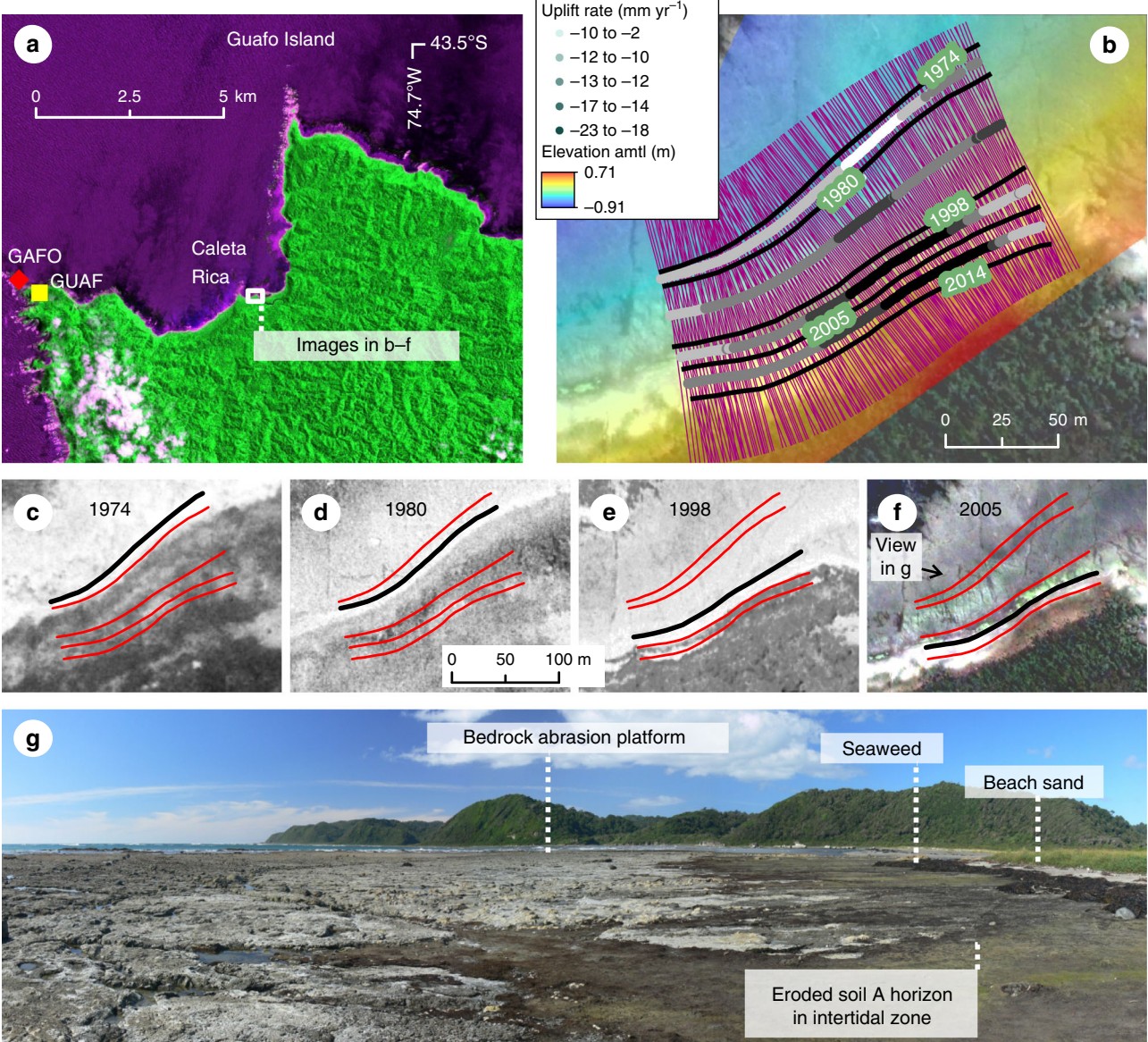

**Fig. 2** Modern relative sea-level (RSL) rise at Guafo. **a** SPOT satellite image showing location of Caleta Rica site and permanent GPS station GUAF and campaign site GAFO. **b** RSL rise estimated using progressive retreat of the boundary between beach sediment and bedrock abrasion platform (using aerial images in **c**) and 0.5-m-scale topography (Methods, Supplementary Table 1, Supplementary Figs. 7-9). Gray-coded stripes show local rate estimated from each adjacent shoreline (labeled with the year) along shore-normal profiles (purple lines). Resulting time series in Fig. 5a and Supplementary Fig. 9. amtl-above mean tide level. Quickbird satellite image in the background. **c–f** Co-registered aerial imagery used to map changing position of shoreline. Image sources and details in Supplementary Table 1. Thick black line denotes mapped shoreline in the corresponding image; red lines denote shorelines on the other images (see Supplementary Fig. 7 for larger versions). **g** View of abrasion platform on which post-1960 peaty meadow soil developed. Location in **f**. Photo by D. Melnick. Copyrighted material for this Figure come from **a** CNES SpotImage, **b**, **f** DigitalGlobe, **c** IGM, **d**, **e** SAF, Inc. All Rights Reserved, used with permission under a NERC-BAS educational license and not included in the Creative Commons license for the article

prediction from our decadal rates of land-level change (Fig. 5d and Supplementary Fig. 11). The viscoelastic effect decays with time, and this time will depend on the viscosity structure of the subduction zone, mainly of the oceanic and continental mantle[12,23]. We estimate the viscoelastic response using a time-dependent model[34], and calibrate the viscosity structure with horizontal and vertical velocities estimated from continuous GPS stations in the region[35] using a three-dimensional (3D) finite-element model with realistic lithospheric geometries[36] (Methods, Supplementary Table 2, Supplementary Fig. 11). Model predictions of the temporal evolution of viscoelastic post-seismic deformation suggest fast subsidence at Guafo with rates as high as 100 mm year$^{-1}$ immediately after the 1960 earthquake,

followed by a rapid decay in subsidence rate in the 1960s followed by uplift at ~3–4 mm year$^{-1}$ thereafter (Fig. 5d).

After considering the results of our viscoelastic model that suggest slow mantle relaxation after the second post-earthquake decade, we used a forward model, constrained to the width of the seismogenic zone, to reconstruct the history of interseismic plate locking deduced from land-level changes at Guafo since the mid 1970s. Constraints on the thermal structure of the margin obtained from modeling of heat-flow data along a profile ~100 km north of Guafo suggest the 350 °C isotherm, which marks the downdip limit of interplate locking, is at ~30 km depth[32]. This depth is consistent with the intersection of the slab with the continental Moho inferred from modeling gravity data[36]. We

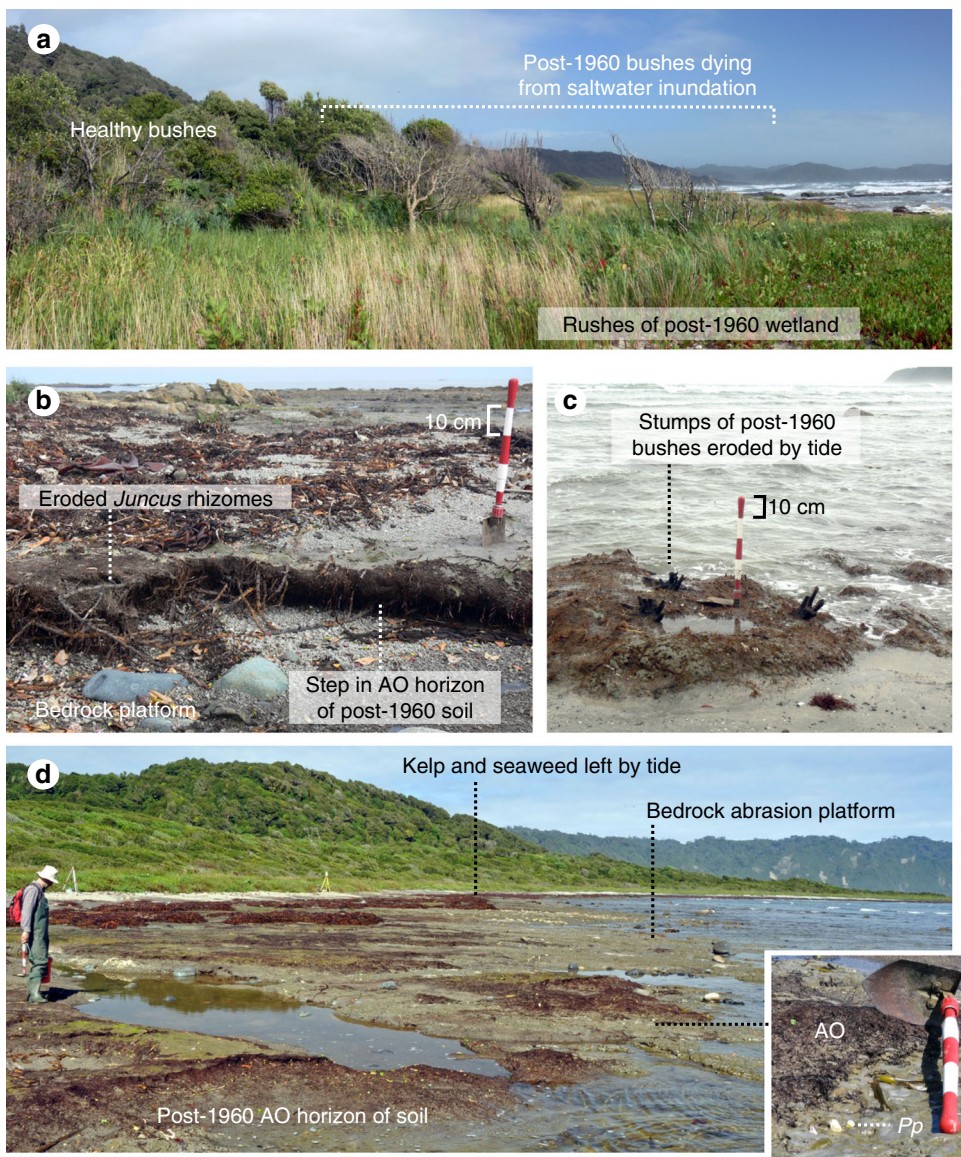

**Fig. 3** Evidence for post-1960 subsidence at Caleta Rica. **a** Post-1960 *Luma apiculata* bushes dying due to saltwater inundation adjacent to healthy rushes (*Juncus* sp.) in post-1960 marsh. Higher bushes in background are healthy. **b** Step in peaty AO horizon of post-1960 soil containing *Juncus* rhizomes. Soil developed directly on abrasion platform of Tertiary bedrock. **c** Stumps of post-1960 *Luma apiculata* bushes in growth position eroded by tides in the present intertidal zone. **d** View of Tertiary bedrock platform with remains of post-1960 wet meadow soil eroded by tides. Inset shows in situ shells of *Petricola patagonica* (*Pp*), a bivalve borer, exhumed from the post-1960 soil by tidal erosion. The shells died following coseismic uplift in 1960 and were covered by the fibrous AO horizon of the soil, which protected them from erosion. Photos by D. Melnick

searched for the width of the locked zone by forward and inverse modeling of horizontal and vertical velocities estimated from three continuous GPS stations using finite-element-based Green's Functions (Methods). The forward models assume that interseismic plate locking is constant over the entire locked zone with linear transitions to creep up- and downdip, searching combinations of up- and downdip locking depth that best reproduce the GPS velocities. We select the inverse model from the trade-off curve between smoothing and model residuals[12,31] (Supplementary Fig. 12). The inverse model reproduces the horizontal velocities better, with full locking conditions between ~10 and ~30 km depth, and then transitionally decreasing to free slip below ~35 km (Fig. 1). The vertical components are better modeled using fully locked conditions between depths of 5 and 35 km (Fig. 4 and Supplementary Fig. 13), with equal root mean

square errors for models using raw GPS velocities and those corrected for post-1960 viscoelastic mantle relaxation. Shallow locking up to ~5 km depth is required to reproduce the fast horizontal motion recorded at GPS station GUAF, and the downdip depth of ~35 km is required to account for the uplift rates observed at inland stations CSTO and MELK. The horizontal velocity at these two latter stations is not well reproduced by the forward modeling suggesting that local-scale processes, such as upper-plate deformation along the intra-arc region[37], may play a secondary role. The inverse model reproduces the horizontal velocities of these two inland stations better. But in order to reproduce both horizontal and vertical components the weight of the vertical GPS velocities needs to be reduced (gray stippled lines in Fig. 4). This probably reflects either rheological complexity and/or upper-plate deformation

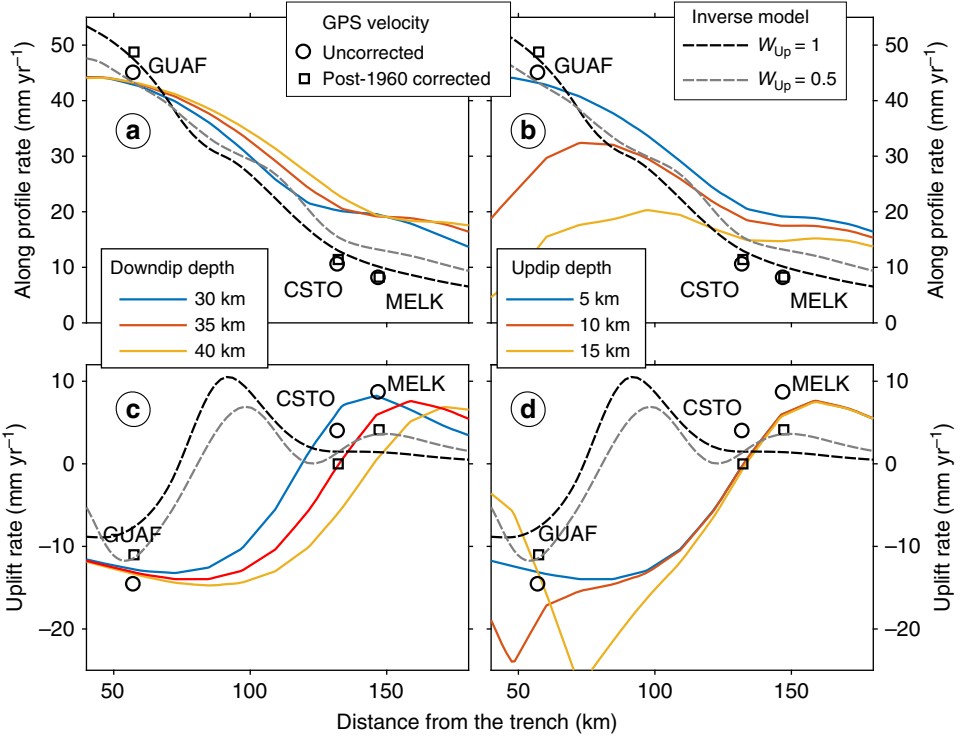

**Fig. 4** Continuous GPS velocities and back-slip forward and inverse models across the south Chile forearc. Velocities projected along the profile line of Fig. 1. Stippled lines in all panels show inverse model results (Methods). $W_{up}$ weight of vertical GPS velocities. **a**, **c** Forward model results for variable downdip locking depths, using an updip depth of 5 km. Variations in the downdip depth mostly influence inland GPS sites CSTO and MELK (location in Fig. 1). Best-fitting depths are consistent with full plate locking down to the intersection of the continental Moho with the slab. **b**, **d** Model results for variable updip locking depths, using a 30-km downdip depth. Note that most variability is in the horizontal component. Shallow locking is required to model the rapid inland velocity at site GUAF. Time series and trajectory models of daily GPS positions and post-seismic model time series in Supplementary Figs. 10 and 11, respectively

across the inland sector. For the purpose of our study, both models yield the same results: full interseismic plate locking below Guafo since 2008.

By assuming that the width of the locked zone has been constant in time, we infer changes in the degree of interseismic plate locking from our history of land-level change at Guafo through forward modeling the same Green's Functions. Our results indicate interseismic plate locking reached ~65% about two decades or less after the earthquake (before 1977) and then increased slowly to ~100% in the following two decades (before 2010) (Fig. 5d). If we include the viscoelastic contribution of the relaxing mantle, interseismic plate locking reached ~70% within the first two decades and ~100% by 2005. These results imply an increase in locking with subdued viscoelastic relaxation several decades after the earthquake.

## Discussion

The increasing rate of subsidence at Guafo following the 1960 earthquake may be the product of three main processes: viscoelastic relaxation of the mantle; shallow afterslip; and interseismic plate locking. Viscoelastic mantle relaxation and afterslip were used to explain the fast rates of post-seismic subsidence documented by GPS following the 2004 Sumatra[38] and 2010 Maule[19] earthquakes at islands in a similar near-trench position to Guafo. Our viscoelastic model predicts similar fast subsidence rates, which decay rapidly in the first post-earthquake decade (Supplementary Fig. 11). The ~10-year decay time is similar to that observed in tide-gauge data from Puerto Montt[39] (projected to ~80 km inland of Guafo), but the duration of decay is much quicker (several decades to a century

shorter) than previous estimates that were based only on horizontal GPS velocities measured between 1994 and 2010[12,40,41]. Our results are another example of how constraining viscoelastic models with horizontal and vertical components of deformation are important in deciphering mantle viscosity structure from geodetic data[34,42].

Afterslip commonly follows great subduction earthquakes[23]. Although we cannot quantify the timescale of this process following the 1960 earthquake, we expect its duration to be ~2–10 years, as found after the 1964 Alaska (M9.2), 2004 Sumatra (M9.2), and 2010 Maule (M8.8) events[43–45]. Therefore, afterslip is unlikely to bias our history of interseismic plate locking at Guafo, which begins over a decade after the 1960 earthquake. If we consider interseismic plate locking to be zero immediately after the earthquake, our results suggest interseismic plate locking was re-established beneath Guafo during the two decades following the earthquake, and then slowly reached full locking three decades thereafter.

The onset of full interseismic plate locking at Guafo is similar to that inferred ~5 decades after the 1835 earthquake at Isla Santa María, ~700 km to the north, using resurveyed nautical charts[19]. As interseismic plate locking obtained over the entire 1960 rupture zone between 2002 and 2010 shows a highly heterogeneous spatial pattern[12], we anticipate differences in the times of re-establishment of interseismic plate locking along this rupture zone and for other great ruptures at other subduction zones. Histories of RSL reconstructed from Sumatran microatolls show pronounced changes in decadal rates, which have been attributed to changes in the locus of interseismic plate locking in time and space[18]. Modeling such data predicts onset of interseismic plate locking at depths > 45 km, which are >10 km into the mantle

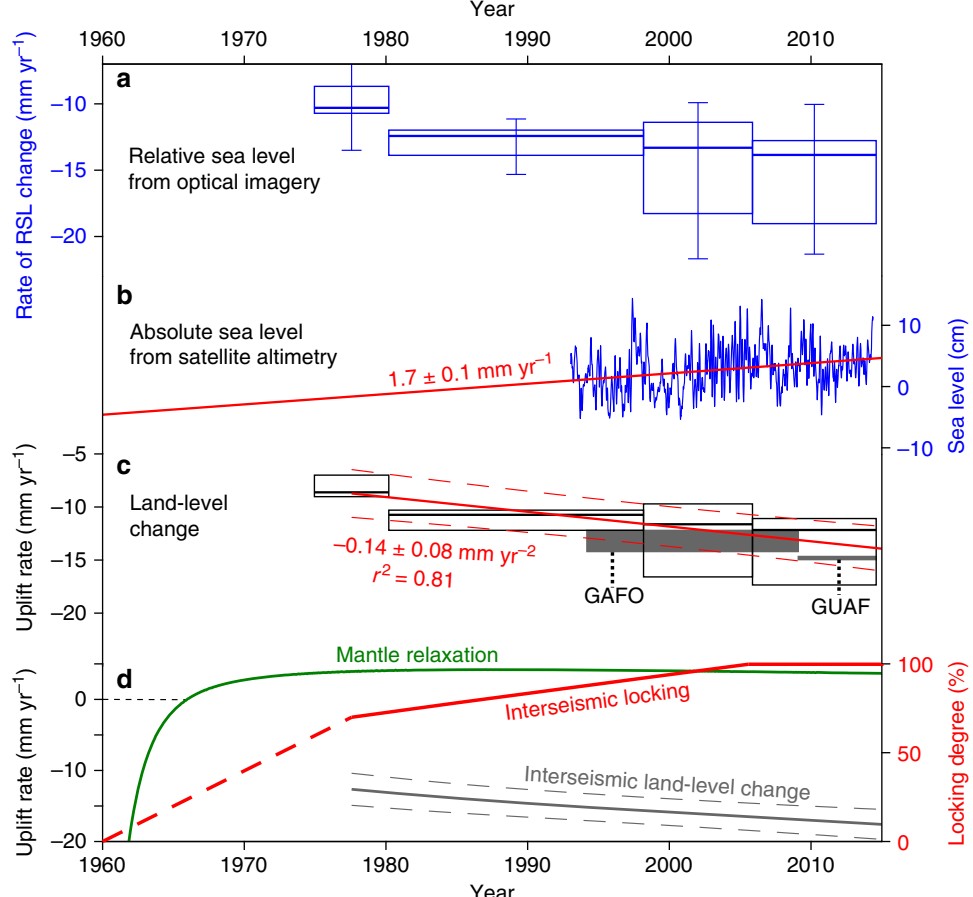

**Fig. 5** Measured and estimated relative sea-level (RSL) and land-level changes at Guafo after the 1960 earthquake. **a** Box-and-whisker plots (center line, median; box limits, upper and lower quartiles; whiskers, 5th and 95th percentiles) showing RSL rise estimated from images in Fig. 2 (Supplementary Fig. 7). **b** Absolute sea level at Guafo from 1992 to 2015 multimission satellite altimetry obtained from AVISO site (French national space agency). Red line shows linear rate. **c** Land-level changes estimated by subtracting relative from absolute sea levels. Red lines show linear acceleration with $2\sigma$ range, estimated from images and uplift rates from labeled GPS stations. **d** History of interseismic plate locking. Green line shows uplift rate at Guafo from viscoelastic mantle relaxation model (Methods); gray line shows rate inferred from interseismic locking obtained by subtracting mantle relaxation from land-level change. Solid red line shows locking rate expressed as fraction of plate convergence, estimated using up- and downdip limits at depths of 5 and 35 km, respectively (Methods, Fig. 4). Stippled red line shows inferred interseismic locking between 1960 and our earliest imagery measurements. The line has been truncated at the full plate locking rate after 2005

wedge, where stable sliding is predicted by the mechanical and thermal boundary conditions[23,32,46]. Because Guafo data cannot resolve spatial variations in interseismic plate locking, we assume that the re-establishment occurred as a gradual change in locking rate.

Temporal variations in interseismic plate locking, like those we model after the 1960 earthquake at Guafo, may be related to the post-earthquake mechanical evolution of the megathrust. Fault healing[47] is the process by which the megathrust regains the strength necessary to transfer stresses across the plate boundary after damage by slip during a great earthquake. It has been proposed that fluid circulation and silica precipitation arising from slab dehydration and mantle serpentinization influence the deformation mode in the transition zone between stable sliding and interplate locking[48,49]. Fluid pressure has been inversely correlated with interseismic plate locking at the northern limit of the 1960 rupture[50], and thus the post-seismic evolution of fault-zone hydrology may be reflected in spatiotemporal variations of interseismic plate locking. Our inferences that interseismic plate locking at Guafo was re-established in less than two decades and then increased more slowly thereafter may reflect the timescales of fault healing processes after the 1960 earthquake. Our reconstruction of the transient rebuilding of interseismic plate locking

following this giant rupture on a decadal scale has implications for the development of time-dependent seismic hazard models based on geodetic data at this and other subduction zones. For example, the hazard in regions identified as highly locked based on modeling of GPS data might be overestimated if the degree of locking is linearly extrapolated back to the time of the previous earthquake. Such overestimation increases if the timing of the previous earthquake is incorrectly interpreted from ambiguous interpretations of historical chronicles[51] or from poorly constrained slip models[52]. Our results bear on interpreting the timescales of earthquake-cycle processes along subduction zones, in particular on the onset of seismogenic loading processes with implications for the use of geodetic data to monitor active megathrusts.

## Methods

**Estimating land-level changes from historical imagery**. We derive RSL changes from historical aerial imagery at Caleta Rica, whose shoreline is characterized by a low relief abrasion platform of homogeneous lithology and the most protected sand beach of the island, making it the best site for our analysis on the island. We first georeferenced a Quickbird satellite image (0.6-m resolution) to ground control points collected with a GNSS differential GPS (Leica 1200). The Quickbird image was used as master scene to co-register the 1974, 1980, and 1998 air photos using the same control points located exclusively at the intersection of fractures in

bedrock platforms (details in Supplementary Table 1). We define the shoreline as the contact between the sandy beach and bedrock platform, evident in the imagery by light and dark tones, respectively. This contact is clearly distinguishable from the different texture between mottled abrasion platform and smooth sandy beach (see insets in Supplementary Fig. 7). The shoreline lies on the bedrock and is therefore not affected by sedimentary processes such as sand accretion during storms. We obtained an along-strike mean shoreline by averaging all the shorelines mapped on the co-registered images; in this way we avoid any bias introduced by using the shoreline from a particular image. Because of the difficulty in estimating the accuracy of our shoreline mapping at a single point, we calculated the horizontal shift between shorelines from image pairs along 400 profiles oriented normal to the along-strike mean shoreline, spaced at 0.5-m intervals. This approach averages out the processes responsible for lateral variability in shoreline development such as drift currents, local changes in bedrock resistance and water depth, and proximity to streams. The topography of the bedrock abrasion platform, surveyed with differential GPS, was used to convert horizontal shoreline shifts to vertical RSL change. In order to reduce the noise associated with the high-frequency rugosity of the platform introduced mostly by differences in the lithology of the Miocene turbiditic bedrock we fitted a surface (Supplementary Fig. 8a) using local linear regression (Matlab lowess function). Residuals of the surface fit follow a Gaussian distribution with a standard deviation of 7 cm (Supplementary Fig. 8b). Because of the low mean slope of the platform ($0.52 \pm 0.28°$), uncertainties in horizontal position are largely insensitive to the estimated changes in RSL. In order to convert RSL change to land-level change, we estimated a mean absolute sea-level trend from multimission satellite altimeter products between 1992 and 2015 (SSALTO/DUACS) available from the AVISO (Archiving, Validation and Interpretation of Satellite Oceanographic data) web site of the French national space agency (www.aviso.altimetry.fr). We subtracted this mean rate from our estimates of RSL changes to obtain land-level changes (Fig. 5), following ref. [53].

**GPS data and trajectory models.** In 2009, we surveyed the campaign GPS benchmark GAFO, installed by the CAP network[54] in 1994, during 4 full days. CAP data were made available by UNAVCO (www.unavco.org). In 2009 we installed the continuous GPS site GUAF at the Guafo lighthouse and MELK at the Melinka harbor, both hosted by the Chilean Navy. The GUAF antenna is at the top of a 123-m-high hill bounded by steep cliffs providing an excellent sky. Station CAST is a merging from pre-2010 data from station CSTR installed by the Chilean Seismological Center (www.sismologia.cl) and post-2010 data from site BN20 (www.catastro.cl), located 1 km apart. The merging is accounted for by an additional step in the heaviside function (see below). The GPS data were processed as part of the IPOC Network (http://www.ipoc-network.org/) using the Earth Parameter and Orbit System (EPOS) software[55] in the IGS08 reference frame[56]. IGS08 Phase center variations and FES2004 ocean tide loading were used with hourly tropospheric wet zenith delays estimated as random-walk parameters and Vienna mapping functions in a grid file database. The reprocessed GNSS precise satellite orbit and clock products are generated together with station coordinates by EPOS and combined with IGS products in order to reduce the impact in estimating Earth rotation parameters. We used the linear trajectory method[35,57,58] to model $x(t)$, the GPS daily position time series in the East, North, and Up components as:

$$x(t) = \sum_{i=1}^{n_p+1} A_i(t - t_R)^{i-1} + \sum_{j=1}^{nj} B_j H(t - t_j) + \sum_{k=1}^{2}\left[C\sin\left(\frac{2\pi}{\tau_k}t\right) + D\cos\left(\frac{2\pi}{\tau_k}t\right)\right]$$
$$+ E\log\left(1 + \Delta t - t_{eq}/T\right)$$
$$(1)$$

where $A$ is the coefficient of polynomial functions of $n_p$ maximum power (we used $n_p = 1$), $t_R$ is a reference time defined as $t_0$, $B$ are the coefficients of $H$ heaviside jumps to simulate earthquakes and non-tectonic effects, $C$ and $D$ are the coefficients of a truncated Fourier series to account for seasonal variations mostly induced by the hydrological cycle (we used $\tau = 1$ year for annual and $\tau = 0.5$ year for semi-annual periods), and $E$ is the coefficient of the transient post-seismic logarithmic component where $t_{eq}$ is the time of the Maule earthquake (27 February 2010), we used $T = 0.1$, the constant determining the timescale of the logarithmic transient following ref. [35]. Heaviside jumps were included for earthquakes from the National Earthquake Information Catalogue of the US Geological Survey located at $d \le 10^{(0.5 \times mag - 0.8)}$, where $d$ is the distance between the epicenter and the GPS station and mag is the earthquake moment magnitude.

**Viscoelastic post-seismic modeling.** All numerical simulations in this study are solved with the finite element software PyLith[59]. The two-dimensional viscoelastic model incorporates the curved geophysically constrained geometry of the trench-normal profile (location in Fig. 1) across the subduction zone[36]. The model consists of elastic oceanic and continental plates overlying viscoelastic oceanic and continental asthenosphere units. We specified a Young's modulus of 100, 120, and 160 GPa, for the continental, oceanic, and mantle layers, respectively[12]. The Poisson's ratio was set to 0.265 and 0.30 for the continental and oceanic crust, respectively. The thickness of the oceanic plate was set to 30 km. We impose the coseismic slip distribution of the 1960 earthquake[60] along the fault interface and simulate post-

seismic motions by relaxing the viscoelastic mantle during the 60-year observation period. The east and west boundaries and the base of the problem domain are held fixed. Following the strategy of ref. [34], we use the current GPS velocities to constrain the laterally heterogeneous viscosity structure (Supplementary Table 2). Two steps are involved. First, we use a homogeneous model to find the optimal viscosity at each station that best explains the observed surface velocity. The constrained viscosity represents the averaged viscosity throughout the simulated time. Then, we construct a lateral heterogeneous viscosity structure in the asthenosphere and use a forward model to predict the preferred displacement time series at the GPS stations. The obtained viscosities are well within estimates from previous studies in Sumatra, Japan, and south Chile[61,62].

**Back-slip interseismic model.** Linear velocities were obtained from the trajectory models of GPS time series and referred to a stable South American reference frame by applying an Euler pole rotation inverted from velocities of stations in the stable continental interior before the 2010 Maule earthquake (location: 21.4°S, 125.2°W; rotation rate: 0.12° Myr⁻¹). Interseismic deformation has been numerically simulated using the back-slip approach, which involves dislocations of locked areas, rather than actual forward slip on surrounding regions. We estimated back-slip displacements (dip-slip) along a curved fault by using finite element method-generated elastic Green's Functions[63], estimated using a 3D model of the margin based on geometries constrained by geophysical data[36]. In order to estimate the modern degree of plate locking below Guafo Island, we performed two different modeling experiments: (1) we assumed full locking conditions below Guafo Island, as deduced from a regional model based on GPS velocities[12], searched for the best combination of up- and downdip locking depths (Fig. 4) using the horizontal and vertical components of continuous GPS stations; and (2) we inverted for locking using the same Green's Functions by searching for the smoothing parameter that controls the Laplacian matrix of the slip distribution from the trade-off curve between misfit and slip roughness[1,12,31]. Results of interseismic locking rate from the inverse model are shown in Fig. 1 and predicted velocities for both models in Fig. 4; the trade-off curve is shown in Supplementary Fig. 12. Based on the best-fitting up- and downdip depth limits for interplate locking, we used the Green's Functions to forward model the history of locking degree that best reproduced the history of land-level changes estimated at Guafo since the 1970s (Fig. 5d).

**Code availability.** Numerical simulations were calculated in Pylith, which is available on the Computational Infrastructure for Geodynamics web site (https://geodynamics.org/cig/software/pylith/). Codes developed in this study to estimate interseismic locking and relative sea-level change are available from the corresponding author upon reasonable request.

**Data availability.** Historical air photos that support the findings of this study are available from the corresponding commercial sources in Supplementary Table 1. Detailed topography is available from the corresponding author upon request. The GPS data are available from UNAVCO (www.unavco.org), the Chilean Seismological Center (www.sismologia.cl), Ministerio de Bienes Nacionales (www.catastro.cl), and for stations MELK and GUAF from the corresponding author upon reasonable request.

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

## Acknowledgements

We acknowledge financial support from the Millennium Nucleus The Seismic Cycle Along Subduction Zones funded by the Millennium Scientific Initiative (ICM) of the

Chilean Government grant NC160025, Chilean National Fund for Development of Science and Technology (FONDECYT) grants 1150321 and 1181479, German Science Foundation (DFG) grants ME 3157/4-2 and MO 2310/1-1, and the U.S. National Science Foundation (NSF) grants RAPID EAR-1036057. R.W. and A.N. are funded by the Earthquake Hazards Program of the U.S. Geological Survey. We thank the Chilean Navy for hosting GPS stations at Isla Guafo lighthouse and Melinka harbor.

## Author contributions

D.M. conceived the project, analyzed the imagery, and wrote the paper. D.M. and R.W. surveyed campaign GPS. D.M., J.J.-M., and J.C.B. installed permanent GPS stations. Z.D. processed GPS data. M.M., S.L., and D.M. designed and conducted the modeling experiments. M.C., R.W., J.J.-M., A.N., and D.M. participated in fieldwork surveying geologic markers of post-1960 subsidence and contributed to data interpretation and writing of the paper.

## Additional information

**Competing interests:** The authors declare no competing interests.

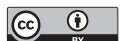 ns license, unless indicated otherwise in a credit line to the material. If material is not included in the article's Creative Commons license and your intended use is not permitted by statutory regulation or exceeds the permitted use, you will need to obtain permission directly from the copyright holder. To view a copy of this license, visit http://creativecommons.org/licenses/by/4.0/.

