## [Peer Review File · Nature Communications]

Reviewers' comments:

Reviewer #1 (Remarks to the Author):

This paper presents a very interesting study showing the slow relocking of the subduction interface in the decades following the 1960 great Chile megathrust earthquake. The authors use an original combination of nautical charts, optical imagery, bathymetric survey and GPS analyses to reconstruct 40 years of vertical movements in Guafo island, located above the locked subduction interface. These vertical movements are then compared to models of post-seismic relaxation and interseismic locking, showing that interplate locking increased to 70% in the decade following the earthquake, and then gradually to 100% by 2005. Those results provide critical constraints on the reloading mechanisms of the subduction interface after a great earthquake, that have important impact on our understanding of the mechanisms at stake on the subduction interface (such as the relation between visco-elastic relaxation, afterslip, fault healing etc...), and on the seismic hazard assessment. This study is of great interest for a broad community of geophysicists and beyond. I therefore recommend publication in Nature Communications.

In addition to the detailed comments listed by line number below, there are a few main points that I suggest to be addressed before publication:

- The modelling of interseismic locking rate is not satisfactory as it shows a huge misfit to the horizontal GPS data. The authors invoke upper plate faulting, but upper plate faulting can not only affect horizontal but also vertical movements. In addition it is usually much smaller than the movements generated by the seismic cycle on the subduction zone. It does not make sense to try to fit a change of $\sim 1\text{cm/yr}$ in subsidence rate, when the model misfits the horizontal rate by $\sim 1\text{cm/yr}$ as well...
- Is it possible to extract from the GPS time series any change in rate during the last decade that would fit with the long-term one? Or is the present-day rate constant? This is probably worth investigating.
- Please explain better how the error bars for subsidence rates have been computed. The authors should also discuss more the misfit between GPS and imagery-derived rates.
- The authors discuss in great details the effect of visco-elastic mantle relaxation, but probably underestimate the effect of afterslip. Guafo island is located above the subduction interface. It is well known that in near field, close to the subduction interface, the effect of mantle visco-elastic relaxation is way smaller than in far field. Instead, afterslip or locking of the plate interface has a strong effect on local near field deformation. This the authors should discuss further.
- The authors should probably refer to other studies showing transient mechanisms at subduction zones, such as post-seismic relocking of preseismic decoupling (e.g. Mavrommatis et al. 2014, Yokota and Koketsu 2015, Socquet et al. 2017, Remy et al. 2016 etc...). In general the authors should be careful to cite works that have been done by other groups in the area and beyond.

Detailed comments:

l28 you may also want to cite the synthesis paper by Metois et al. (2016) that quantifies the link between interseismic and co-seismic.

l42 & l50-52 you may cite here recent studies showing that interseismic locking can change over months / decades before a major megathrust earthquake (e.g. Yokota & Koketsu 2015, Mavrommatis et al. 2014, Socquet et al. 2017 etc..)

l44 please define RSL

l49 post-seismic mantle relaxation is likely to last several decades and affect a wide continental area (e.g. Wang et al. 2012)

l63-65 Please mention the period this sentence refers to.

l73 deformation -> displacements, is -> are

l125 Could you specify that the shoreline mapped represents? Is it the upper limit of the highest tide

?

l131-133 The authors should rather say that their 1998-2005 & 2005-2017 rates derived from aerial imagery fit well with the GPS vertical rate.

l135 an increasing rate of -> an increasing rate from

l301-302 The authors "define the shoreline as the contact between the sandy beach and bedrock platform, evident in the imagery by light and dark tones, respectively." This limit is actually not obvious to map from aerial photos, in particular in 1974.

l302-304 "The shoreline lies on the bedrock and is therefore not affected by sedimentary processes such as sand accretion during storms." I don't understand why the authors say this.

l305-307 This is not clear.

l150-151 "in the second post-earthquake and later decades" : clarify this sentence that sounds weird to me.

l155 "suggest the 350°C isotherm, which marks the downdip limit of interplate locking is at ~30 km depth" : reformulate, or add a comma after "downdip limit of interplate locking"

l159: instead of "modern" use "present-day" or "current" and give into brackets the time span on which the GPS velocity field is computed

l165-167 and figure 4: I am really puzzled by the misfit between horizontal velocities and the model. Not sure that it can be explained only by local-scale processes such as upper plate faulting. Usually upper plate faulting account for strain that are an order of magnitude smaller than those generated by the seismic cycle on the subduction zone. It does not make sense that the authors try to fit the vertical deformation, and accept having such a large misfit in horizontal velocities. It is well known that the vertical component is probably the most difficult to model given the high complexity of the processes involved (including non-linear viscoelastic relaxation...). The horizontal component is instead usually more easy to fit. So the author should provide more precise explanation and further evidence if they actually believe that this huge misfit (~1cm/yr) is generated by local faulting. Would a shallower downdip depth provide a better fit?

l177 In fact, viscoelastic relaxation of the mantle predicts decreasing subsidence. The authors should also probably further investigate the effect of shallow afterslip on vertical motion. This is not presented on this paper and is likely to have an inverted effect to the one of interseismic locking. In other words, while there is afterslip, the coupling is low since the plate interface is creeping, and when afterslip decreases, the plate interface is starting to be healing, generating an increased locking.

l186-187: "Our results are another example of how constraining viscoelastic models with horizontal and vertical components of deformation are important in deciphering mantle viscosity structure from geodetic data^{29,38}." Given the location of Gafu island right above the subduction seismogenic zone, I don't think that the main post-seismic effect here is the viscoelastic relaxation (e.g. Wang et al. 2012). Afterslip may instead play a major role in the years following the earthquake that is in my opinion not enough discussed nor modelled in this paper, the other major effect being the slow increase in interseismic coupling associated with e.g. healing mechanisms. So I think that the main reason why the decay time is much shorter in Gafu than inland is because these different locations are affected by different post-seismic mechanisms. I would delete this sentence, and discuss the relative contribution of afterslip on the subduction interface mainly responsible for the deformation in Gafu (in near field) versus viscoelastic mantle relaxation responsible for large time and space scale processes.

l193-195: You may want to cite here other examples of observed progressive interseismic reloading following a subduction earthquake such as Remy et al. 2016 who documented this effect following Pisco 2007 earthquake.

Figure 5:

a/ The way error bars on RSL rate change are computed are not clear to me, and would probably require further details in the method text.

c/ your estimated linear acceleration of subsidence does not fit the present-day GPS rate at GUAF

station, which is not satisfactory to me, given the accuracy of GPS stations compared to the one estimated from imagery. You should revisit the weights / errors included in your data set. Also, given that GUAQ GPS station has now been measuring for more than five years now, can you try to look in your time series whether or not you find an acceleration that would fit with your long term one. You could for example follow a procedure similar to the one of Socquet et al. (2017) in order to look for potential changes in velocity.

Mavrommatis, A. P., P. Segall, and K. M. Johnson (2014), A decadal-scale deformation transient prior to the 2011 Mw 9.0 Tohoku-oki earthquake, *Geophys. Res. Lett.*, 41, 4486–4494, doi:10.1002/2014GL060139.

Metois, M., C. Vigny, and A. Socquet (2016), Interseismic coupling, megathrust earthquakes and seismic swarms along the Chilean subduction zone (38°–18°S), *Pure Appl. Geophys.*, 194(3), 1283–1294, doi:10.1007/s00024-016-1280-5.

Remy, D., et al. (2016), Postseismic relocking of the subduction megathrust following the 2007 Pisco, Peru, earthquake, *J. Geophys. Res. Solid Earth*, 121, doi:10.1002/2015JB012417.

Socquet, A., et al. (2017), An 8 month slow slip event triggers progressive nucleation of the 2014 Chile megathrust, *Geophys. Res. Lett.*, 44, doi:10.1002/2017GL073023.

Wang, K., Hu, Y., & He, J. (2012). Deformation cycles of subduction earthquakes in a viscoelastic Earth. *Nature*, 484(7394), 327–332.

Yokota, Y. and Koketsu, K. A very long-term transient event preceding the 2011 Tohoku earthquake. *Nat. Commun.* 6:5934 doi: 10.1038/ncomms6934 (2015).

Reviewer #2 (Remarks to the Author):

Review

Nature Communication

Title: Back to full interseismic plate locking decades after the giant 1960 Chile earthquake

The whole concept of interpreting the variation in the uplift rate in terms of estimating the locking extent on the main thrust zone is good. Although there are several assumptions, but with the available limited data and information about the past, this is the best, one can do. I suggest that the authors clearly mention this in the manuscript.

Estimating Postseismic Deformation and its dominance after an earthquake is one of the biggest issues. In the absence of the data on the postseismic deformation, it is very difficult to convince that postseismic deformation (uplift/subsidence) did not continue beyond a few years time after the 1960 earthquake (M9.5). We know that in case of Sumatra Andaman earthquake (M9.2), the postseismic deformation is very dominating even after 14 years and we hope that it would continue for some more years.

It is possible that a few regions (with a huge rupture length), which could be the regions of low/high seismic release or rheologically different from neighbouring regions, they may show a different behaviour in terms of postseismic deformation. So assuming a simple model for postseismic deformation (viscoelastic relaxation) with rapid decay in uplift rate is a bit overstretch. Moreover, I am not sure whether afterslip occurred during this earthquake and it was only viscoelastic relaxation, which operated. At least these limitations should be mentioned.

While interpreting the GPS measurements, it suggested that authors should show scenarios with locking, as 90%, 80% With these limited data (only 3 sites) we do not know whether the locking is 100% at present or partial.

Also the question remains that how the interseismic uplift rates over 1978-2005 have been estimated. You are assuming that the convergence velocity is unchanged over time and its only the locking extent which is changing. So during 1978-2005 period, the interseismic uplift rate is increasing as the locking is increasing, but then after 2005, when the locking is 100% and is constant, then why the interseismic uplift rate is still increasing. It should also be constant.

Overall, I am fine with the GPS part and expect authors to highlight their limitations.

Whereas, the other portion of the paper describes the land-level change using satellite data.

Lines 114-115: Here you say slow RSL rise post 1960. However, in line 122 you mention rapid post-1960 subsidence. How do you explain this??

Lines 127-128: What was the time frame in which you collected your data? How did you extrapolated your generated data collected during last?

Figure 2: Please put coordinates in figures 2b-c.

Reviewer #3 (Remarks to the Author):

[Editorial Note: See the reviewer comments on the following page]

This work reconstructs the history of plate locking in a segment of the Chilean subduction zone that ruptured during the 1960 M9.5 Chile earthquake. The authors use aerial imagery and GPS measurements (spanning decades) to constrain the evolution of plate locking since the 1960 earthquake. As the authors note, a subduction zone's present day locking (i.e., assuming the locking fraction is constant through time) is often used to infer something about the energy available for future earthquakes. Thus, changes to the locking history over time are critical to the accuracy of these estimates, and are of broad interest to the community.

It appears to me that somewhat similar papers have been published previously (e.g., references 13 and 14), and the manuscript would benefit from plainly stating the advantages of the current study. Loveless and Meade (EPSL, 2016), which examines a smaller time range but also suggests fluid pressure may explain spatiotemporal variations in coupling (i.e., locking) behavior, could also be cited.

Except for my concern about the low viscosity beneath GUAF and its effect on the results, the vast majority of my comments are either minor or related to issues of clarity. (See below)

L. 21 "... of **relative** quiescence..." (In most subduction zones, there are still smaller earthquakes.)

L. 23 Change "and so" to "and thus,"

L. 44 Define RSL

L. 46 Was there some degree of locking, it was just small? i.e., Do you mean "full locking" wasn't established until decades after the earthquake?

L. 87/115/etc. It should be explicitly stated somewhere that interseismic subsidence of Guafo Island contributes to the gradual RSL rise

L. 95 (i.e., the AO horizon)

L. 101 "... supported **by observations**..."

L. 142 I'm surprised by the extremely low viscosity beneath GUAF. What could be causing this low viscosity? (Even in the hydrated mantle wedge, viscosities are expected to be $\sim 10^{18}$ Pa s.) This also suggests a strong lateral gradient in viscosity – is this physically plausible? How sensitive to the viscosity structure (in general) are your results? If you use a higher viscosity, does this affect the rapid decay of viscoelastic effects?

L. 158 "... to creep **in** the up- and down-dip **regions**,..."

L. 182 Indicate where Puerto Montt is located

L. 212 I would consider removing "mantle serpentinization." Presumably, "fluid circulation and silica precipitation" does not **require** the mantle wedge to be serpentinized. If the mantle wedge were fully serpentinized, this would just make **more** fluid available, as dehydrated fluids wouldn't be consumed to further serpentinize the mantle.

Figure 1

- Remove "Index maps" from the caption.

Figure 2

- Clarify the meaning of "SPOT image." What is the color scale?
- The uplift rate colors in the legend (green hues) do not match the colors in the figure itself (gray hues).
- What is "amtl"-above?
- Define the red lines. Does the most landward red line represent present-day?

Figure S5

- Replace "???" with something more descriptive

Reviewer #1 (Remarks to the Author):

This paper presents a very interesting study showing the slow relocking of the subduction interface in the decades following the 1960 great Chile megathrust earthquake. The authors use an original combination of nautical charts, optical imagery, bathymetric survey and GPS analyses to reconstruct 40 years of vertical movements in Guafo island, located above the locked subduction interface. These vertical movements are then compared to models of post-seismic relaxation and interseismic locking, showing that interplate locking increased to 70% in the decade following the earthquake, and then gradually to 100% by 2005. Those results provide critical constraints on the reloading mechanisms of the subduction interface after a great earthquake, that have important impact on our understanding of the mechanisms at stake on the subduction interface (such as the relation between visco-elastic relaxation, afterslip, fault healing etc...), and on the seismic hazard assessment. This study is of great interest for a broad community of geophysicists and beyond. I therefore recommend publication in Nature Communications. In addition to the detailed comments listed by line number below, there are a few main points that I suggest to address before publication:

- The modelling of interseismic locking rate is not satisfactory as it shows a huge misfit to the horizontal GPS data. The authors invoke upper plate faulting, but upper plate faulting can not only affect horizontal but also vertical movements. In addition it is usually much smaller than the movements generated by the seismic cycle on the subduction zone. It does not make sense to try to fit a change of ~ 1 cm/yr in subsidence rate, when the model misfits the horizontal rate by ~ 1 cm/yr as well...

We performed an additional modelling experiment by inverting the GPS velocities. The fit for the horizontal components of the inland stations is indeed better (see the revised version of Fig. 4); however, the results in terms of the temporal evolution of plate locking below Guafo remain unchanged.

- Is it possible to extract from the GPS time series any change in rate during the last decade that would fit with the long-term one? Or is the present-day rate constant? This is probably worth investigating.

Indeed, this was the aim of a previous study (Melnick et al. 2017 GRL) that included a continental-scale network of cGPS stations. However, we found no significant changes at the cGPS stations used in this study.

- Please explain better how the error bars for subsidence rates have been computed. The authors should also discuss more the misfit between GPS and imagery-derived rates.

The procedure for determining error bars using imagery has been described in the methods section. We expanded the description to provide more insight to readers.

- The authors discuss in great details the effect of visco-elastic mantle relaxation, but probably underestimate the effect of afterslip. Guafo island is located above the subduction interface. It is well known that in near field, close to the subduction interface, the effect of mantle visco-elastic relaxation is way smaller than in far field. Instead, afterslip or locking of the plate interface has a strong effect on local near field deformation. The authors should discuss this further.

The first paragraph of the discussion section addresses the potential effect of afterslip. Because of its usual timescale (< 10 years after the earthquake) it is unlikely to affect our observations, which start > 10 years after the earthquake.

- The authors should probably refer to other studies showing transient mechanisms at subduction zones, such as post-seismic relocking of preseismic decoupling (e.g. Mavrommatis et al. 2014, Yokota and Koketsu 2015, Socquet et al. 2017, Remy et al. 2016 etc...). In general the authors should be careful to cite works that have been done by other groups in the area and beyond.

We appreciate the comment and included the suggested reference at the end of the first paragraph. These helped also to provide a better context of our study.

Detailed comments:

I28 you may also want to cite the synthesis paper by Metois et al. (2016) that quantifies the link between interseismic and co-seismic.

We appreciate the comment and included the reference.

I42 & I50-52 you may cite here recent studies showing that interseismic locking can change over months / decades before a major megathrust earthquake (e.g. Yokota & Koketsu 2015, Mavrommatis et al. 2014, Socquet et al. 2017 etc..)

We appreciate the comment and included the first two references, which analysed time series of more than a decade in length, in the context of our study.

I44 please define RSL

We apologize for this mistake; the definition has been included (relative sea level).

I49 post-seismic mantle relaxation is likely to last several decades and affect a wide continental area (e.g. Wang et al. 2012)

Yes, but only if we use homogeneous viscosity and analyse only the horizontal component of GPS velocities, as done by Wang et al. 2012. In our study, we use a more sophisticated model and explore both horizontal and vertical components, with different results. Our shorter timescale (for the fast response) is consistent with tide gauge data at Puerto Montt (Fig. R1; Ding and Lin, 2012 GJI), which is the only data available in the first decade after the earthquake.

Fig. R1. Time series of tide gauge station at Puerto Montt, extracted from Ding and Lin (2012)

I63-65 Please mention the period this sentence refers to.

The revised version includes “as estimated using GPS velocities collected mostly between 2002-2010”.

I73 deformation -> displacements, is -> are

We added “The amplitude of” to clarify the sentence.

I125 Could you specify that the shoreline mapped represents? Is it the upper limit of the highest tide ?

The definition of shoreline had been specified in parentheses (*i.e., limit between dark bedrock abrasion platform and light sandy beach*). Further details can be found in the Methods section and Supplementary

Figures. The shoreline refers to the base of the sandy beach and therefore is not related to tides.

l131-133 The authors should rather say that their 1998-2005 & 2005-2017 rates derived from aerial imagery fit well with the GPS vertical rate.

This is exactly what this sentence means.

l135 an increasing rate of -> an increasing rate from

This has been changed.

l301-302 The authors "define the shoreline as the contact between the sandy beach and bedrock platform, evident in the imagery by light and dark tones, respectively." This limit is actually not obvious to map from aerial photos, in particular in 1974.

This is clearly visible as a change in texture between mottled abrasion platform and smooth sandy beach (see insets in Fig. S7). In order to ease lecture, we refer to these and other features in the methods section.

l302-304 "The shoreline lies on the bedrock and is therefore not affected by sedimentary processes such as sand accretion during storms. " I dont understand why the authors say this.

Because sand accretion during storms occurs along the berm (upper part of the beach).

l305-307 This is not clear.

We rephrase without passive voice to make this sentence clearer. The mean shoreline is used to extract the profiles avoiding the bias introduced by using the shoreline mapped from a particular image.

l150-151 "in the second post-earthquake and later decades" : clarify this sentence that sounds weird to me.

This sentence has been changed to "*after the second post-earthquake decade*".

l155 "suggest the 350oC isotherm, which marks the downdip limit of inter plate locking is at ~30 km depth" : reformulate, or add a coma after "downdip limit of interplate locking"

We added the coma as suggested.

l159: instead of "modern" use "present-day" or "current" and give into brackets the time span on which the GPS velocity field is computed

We removed 'modern'. Details on cGPS data are included in the Methods section and Supp. Figures.

l165-167 and figure 4: I am really puzzled by the misfit between horizontal velocities and the model. Not sure that it can be explained only by local-scale processes such as upper plate faulting. Usually upper plate faulting account for strain that are an order of magnitude smaller than those generated by the seismic cycle on the subduction zone. It does not make sense that the authors try to fit the vertical deformation, and accept having such a large misfit in horizontal velocities. It is well known that the vertical component is probably the most difficult to model given the high complexity of the processes involved (including non-linear viscoelastic relaxation...). The horizontal component is instead usually more easy to fit. So the author should provide more precise explanation and further evidence if they actually believe that this huge misfit (~1cm/yr) is generated by local faulting. Would a shallower downdip depth provide a better fit?

We appreciate this comment and addressed it by inverting the GPS velocities. The fits for the inland stations are indeed slightly better (see the revised version of Fig. 4); however, the results in term of the temporal evolution of plate locking below Guafo remain unchanged.

1177 In fact, viscoelastic relaxation of the mantle predicts decreasing subsidence. The authors should also probably further investigate the effect of shallow afterslip on vertical motion. This is not presented on this paper and is likely to have an inverted effect to the one of interseismic locking. In other words, while there is afterslip, the coupling is low since the plate interface is creeping, and when afterslip decreases, the plate interface is starting to be healing, generating an increased locking.

The reviewer is correct, but as mentioned in the discussion section, the effect of afterslip should have lasted only during the first post-earthquake decade (e.g., Lange et al., 2014 GJI; Qiu et al., 2018 Nature Communications), and therefore is unlikely to affect our results. This reasoning, justified by case studies in other subduction zones, had been included in the discussion section.

1186-187: "Our results are another example of how constraining viscoelastic models with horizontal and vertical components of deformation are important in deciphering mantle viscosity structure from geodetic data^{29,38}." Given the location of Gafu island right above the subduction seismogenic zone, I don't think that the main post-seismic effect here is the viscoelastic relaxation (e.g. Wang et al. 2012). Afterslip may instead play a major role in the years following the earthquake that is in my opinion not enough discussed nor modelled in this paper, the other major effect being the slow increase in interseismic coupling associated with e.g. healing mechanisms. So I think that the main reason why the decay time is much shorter in Gafu than inland is because these different locations are affected by different post-seismic mechanisms. I would delete this sentence, and discuss the relative contribution of afterslip on the subduction interface mainly responsible for the deformation in Gafu (in near field) versus viscoelastic mantle relaxation responsible for large time and space scale processes.

Again, the reviewer is correct, but as mentioned in the discussion section, the effect of afterslip should have lasted only during the first post-earthquake decade (e.g., Lange et al., 2014 GJI; Qiu et al., 2018 Nature Communications), and therefore is unlikely to affect our results. This reasoning, justified by case studies in other subduction zones, had been included in the discussion section.

1193-195: You may want to cite here other examples of observed progressive interseismic relocking following a subduction earthquake such as Remy et al. 2016 who documented this effect following the Pisco 2007 earthquake.

We appreciate the comment but have not included this reference because it is not comparable with the other case studies, due to the rather low magnitude ($M8$) of the Pisco earthquake, and lack of GPS data at islands located directly above the seismogenic zone (such as Gafu).

Figure 5:

a/ The way error bars on RSL rate change are computed are not clear to me, and would probably require further details in the method text.

The methods section describes how error bars are computed. We expanded this section to help readers better understand our novel approach.

c/ your estimated linear acceleration of subsidence does not fit the present-day GPS rate at GUAF station, which is not satisfactory to me, given the accuracy of GPS stations compared to the one estimated from imagery. You should revisit the weights / errors included in your data set.

Because we are looking at multi-decadal changes, the important point here is that the present-day rate at GUAF does fit the RSL rate within error bounds. Please note that the rate at GUAF falls well within the lower red dotted line in Fig. 5c, which represents the 95% confidence interval of the linear regression.

Also, given that GUAF GPS station has now been measuring for more than five years now, can you try to look in your time series whether or not you find an acceleration that would fit with your long term one. You could for example follow a procedure similar to the one of Socquet et al. (2017) in order to look for potential

changes in velocity.

In a previous study (Melnick et al. 2017 GRL), we searched for changes in GPS velocities at regional scale, finding no significant variations at GUAFO.

Reviewer #2 (Remarks to the Author):

Review

Nature Communication

Title: Back to full interseismic plate locking decades after the giant 1960 Chile earthquake

The whole concept of interpreting the variation in the uplift rate in terms of estimating the locking extent on the main thrust zone is good. Although there are several assumptions, but with the available limited data and information about the past, this is the best, one can do. I suggest that the authors clearly mention this in the manuscript.

We appreciate the comment and included a paragraph highlighting the importance of having well established 'background' or 'secular' trends, and a line emphasizing the novelty of our study at the end of the introduction.

Estimating Postseismic Deformation and its dominance after an earthquake is one of the biggest issues. In the absence of the data on the postseismic deformation, it is very difficult to convince that postseismic deformation (uplift/subsidence) did not continue beyond a few years time after the 1960 earthquake (M9.5). We know that in case of Sumatra Andaman earthquake (M9.2), the postseismic deformation is very dominating even after 14 years and we hope that it would continue for some more years.

The reviewer is correct, and in fact our model results suggest the same: that post-seismic deformation continued for about two decades after the 1960 earthquake. This is supported by tide gauge data – the only direct observation at this timescale. Data from earthquakes of similar magnitude in Alaska and Sumatra support our inference.

It is possible that a few regions (with a huge rupture length), which could be the regions of low/high seismic release or rheologically different from neighbouring regions, they may show a different behaviour in terms of postseismic deformation. So assuming a simple model for postseismic deformation (viscoelastic relaxation) with rapid decay in uplift rate is a bit overstretch. Moreover, I am not sure whether afterslip occurred during this earthquake and it was only viscoelastic relaxation, which operated. At least these limitations should be mentioned.

The reviewer is correct in his comments, which both had been included in the manuscript. We accounted for heterogeneous variations in viscosity (rheology) and discussed the potential effect of afterslip.

While interpreting the GPS measurements, it suggested that authors should show scenarios with locking, as 90%, 80% With these limited data (only 3 sites) we do not know whether the locking is 100% at present or partial.

We show that 100% is required to reproduce the fast inland motion (50 mm/yr) and fast subsidence (16 mm/yr) at GUAFO today, both using forward and inverse models. We need to assume a similar spatial distribution while changing the percentage back in time. The change to 90% or 80% suggested by the reviewer only affects the amplitude of velocities, not the spatial distribution, which needs to be fitted by different widths (as shown in Fig. 4).

Also the question remains that how the interseismic uplift rates over 1978-2005 have been estimated. You are assuming that the convergence velocity is unchanged over time and it's only the locking extent which is changing. So during 1978-2005 period, the interseismic uplift rate is increasing as the locking is increasing,

but then after 2005, when the locking is 100% and is constant, then why the interseismic uplift rate is still increasing. It should also be constant.

The reviewer is correct. We truncated the line in Fig. 5d for clarity and because the change in uplift rate at this point falls within the 95% confidence interval of our linear regression. We now mention this clearly in the caption.

Overall, I am fine with the GPS part and expect authors to highlight their limitations.

We have mentioned the limitations of using only the horizontal components of GPS velocities in both postseismic and interseismic models. Technical limitations of GPS have been broadly documented and are beyond the scope of our study.

Whereas, the other portion of the paper describes the land-level change using satellite data.
Lines 114-115: Here you say slow RSL rise post 1960. However, in line 122 you mention rapid post-1960 subsidence. How do you explain this??

This sentence refers to the dramatic and sudden drop in RSL during the 1960 earthquake (3.6 – 4.0 m) and the 'comparatively much slower' RSL rise that followed in the decades thereafter. However, in order to avoid confusions, we removed 'slow' and changed 'rapidly' by 'suddenly'.

Lines 127-128: What was the time frame in which you collected your data? How did you extrapolated your generated data collected during last?

These and other details had been provided in the Methods section.

Figure 2: Please put coordinates in figures 2b-c.

We included reference coordinates to Figure 2a, but for clarity prefer no to include them in 2b-c. Larger versions of these maps with more interpreted features and coordinates are included in Figure S7 .

Reviewer #3 (Remarks to the Author):

This work reconstructs the history of plate locking in a segment of the Chilean subduction zone that ruptured during the 1960 M9.5 Chile earthquake. The authors use aerial imagery and GPS measurements (spanning decades) to constrain the evolution of plate locking since the 1960 earthquake. As the authors note, a subduction zone's present day locking (i.e., assuming the locking fraction is constant through time) is often used to infer something about the energy available for future earthquakes. Thus, changes to the locking history over time are critical to the accuracy of these estimates, and are of broad interest to the community.

It appears to me that somewhat similar papers have been published previously (e.g., references 13 and 14), and the manuscript would benefit from plainly stating the advantages of the current study. Loveless and Meade (EPSL, 2016), which examines a smaller time range but also suggests fluid pressure may explain spatiotemporal variations in coupling (i.e., locking) behavior, could also be cited.

We included this and other references suggested by Reviewer #1 in the second paragraph of the introduction section.

Except for my concern about the low viscosity beneath GUAF and its effect on the results, the vast majority of my comments are either minor or related to issues of clarity. (See below)

L. 21 "... of **relative** quiescence..." (In most subduction zones, there are still smaller earthquakes.)

Exactly this is the reason why we included 'relative'.

L. 23 Change “and so” to “and thus,”

This has been changed.

L. 44 Define RSL

We apologize for this mistake and have included the definition (relative sea level).

L. 46 Was there some degree of locking, it was just small? i.e., Do you mean “full locking” wasn’t established until decades after the earthquake?

As stated in the sentence, we mean that “full locking” was not established until about the year 2005.

L. 87/115/etc. It should be explicitly stated somewhere that interseismic subsidence of Guafo Island contributes to the gradual RSL rise

We refer to this in the discussion section, as we prefer not to mix observations (RSL) with processes (interseismic) in the results section.

L. 95 (i.e., the AO horizon)

This suggestion has been included

L. 101 “... supported **by observations**...”

This suggestion has been included

L. 142 I’m surprised by the extremely low viscosity beneath GUAF. What could be causing this low viscosity? (Even in the hydrated mantle wedge, viscosities are expected to be $\sim 10^{18}$ Pa s.) This also suggests a strong lateral gradient in viscosity – is this physically plausible? How sensitive to the viscosity structure (in general) are your results? If you use a higher viscosity, does this affect the rapid decay of viscoelastic effects?

Here, we refer to time-dependent transient viscosity, not long-term ‘steady-state’ viscosity, which is what we presume the reviewer is referring to. Numerical models show that viscosities are timescale dependent (e.g., Sobolev and Muldashev, 2017 G3), and relatively low values $\sim 10^{18}$ Pa s are commonly needed to reproduce postseismic geodetic data. For example, Qiu et al. (2018 Nature Communications) using decadal-scale GPS data at Sumatra obtained transient viscosities with strong spatial variability (as in our study) that range from 10^{17} to 10^{21} Pa s.

L. 158 “... to creep **in** the up- and down-dip **regions**,...”

This suggestion has been included.

L. 182 Indicate where Puerto Montt is located

We added a square to Fig. 1a showing the location.

L. 212 I would consider removing “mantle serpentinization.” Presumably, “fluid circulation and silica precipitation” does not **require** the mantle wedge to be serpentinized. If the mantle wedge were fully serpentinized, this would just make **more** fluid available, as dehydrated fluids wouldn’t be consumed to further serpentinize the mantle.

This suggestion has been included.

Figure 1

- Remove “Index maps” from the caption.

This suggestion has been included.

Figure 2

- Clarify the meaning of “SPOT image.” What is the color scale?

SPOT holds for: “Satellite Pour l’Observation de la Terre”. We leave it to the editorial staff to decide if it needs to be included in the legend; however, we added “satellite” after SPOT to clarify.

- The uplift rate colors in the legend (green hues) do not match the colors in the figure itself (gray hues).
- What is “amtl”-above?

“amtl - above mean tide level”. We added spaces to the hyphen for clarity. The colours do fit the figure as they were generated using the same software.

- Define the red lines. Does the most landward red line represent present-day?

We added: “Red lines are shorelines of the other images (see Fig. S7 for larger versions)”.

Figure S5

Replace “??” with something more descriptive

We apologize for this typo.

References:

- Qiu, Q., Moore, J.D.P., Barbot, S., Feng, L., Hill, E.M., 2018. Transient rheology of the Sumatran mantle wedge revealed by a decade of great earthquakes. *Nature Communications* 9, 995.
- Lange, D., Bedford, J., Moreno, M., Tilmann, F., Baez, J., Bevis, M., and Krueger, F., 2014, Comparison of postseismic afterslip models with aftershock seismicity for three subduction-zone earthquakes: Nias 2005, Maule 2010 and Tohoku 2011: *Geophysical Journal International*, v. 199, no. 2, p. 784-799.
- Melnick, D., Moreno, M., Quinteros, J., Baez, J. C., Deng, Z., Li, S., and Oncken, O., 2017, The super-interseismic phase of the megathrust earthquake cycle in Chile: *Geophysical Research Letters*, v. 44, no. 2, p. 784-791.
- Sun, T., Wang, K., Iinuma, T., Hino, R., He, J., Fujimoto, H., Kido, M., Osada, Y., Miura, S., Ohta, Y., Hu, Y., 2014. Prevalence of viscoelastic relaxation after the 2011 Tohoku-oki earthquake. *Nature* 514, 84.
- Sobolev, S. V., and Muldashev, I. A., 2017, Modeling Seismic Cycles of Great Megathrust Earthquakes Across the Scales With Focus at Postseismic Phase: *Geochemistry, Geophysics, Geosystems*, v. 18, no. 12, p. 4387-4408.
- Wang, K., Hu, Y., and He, J., 2012, Deformation cycles of subduction earthquakes in a viscoelastic Earth: *Nature*, v. 484, no. 7394, p. 327-332.

Reviewers' comments:

Reviewer #1 (Remarks to the Author):

Overall the authors have addressed most of the comments raised by the reviewers. I still have two main comments relative to the assumption made for inter and post seismic modelling, and the way results are presented.

Interseismic loading:

It seems that authors have performed a new model so that GPS data are reasonably fit. However, new figure 4 is not clearly explained. It seems that the authors invert for the interseismic coupling. To what upper and lower locking depths does the inverted model correspond to? This should be mentioned in the text. I also suggest that the authors add the iso-depth contours of the slab in the figure 1.

I am still quite puzzled by the difference between the forward models and the inverse model in the predicted horizontal components... What if the author propose a forward model with a much shallower locking depth (e.g. 15 or 20 km). This would fit the data much better, and I think it would also correspond to the results of the inversion. I think it is really strange that the authors still chose to show models with locking depths ranging from 30 to 40 km, when the data seem to show something much more shallow. Why do the authors do this? Do they want to avoid showing such a shallow locking? If so why? If the authors have an a priori, it should be explained more clearly... The way it is currently presented indirectly suggests to the reader that the locking is full between 5 and 30 km, which is not what is shown by the data.

Post-seismic:

The authors discard the possibility of long-term afterslip, but they instead use a very low transient viscosity, which is meant to fit relatively short relaxation time. I am not sure that one can discriminate between both phenomena actually... Long term GPS transient have been observed and modelled as long lasting slow slip on the subduction interface. I do not understand why the authors rule out the possibility of such long-lasting afterslip, since no earthquake of such large magnitude has ever been measured... I think this is more a principle position from the authors, than any data driven observable...

These remarks however do not change the main conclusions of the paper, which novelty is to be recognized. I therefore recommend publication once my suggestions to clearly show the depth of the inverted boundaries of the interseismic coupling have been taken in consideration.

Reviewer #2 (Remarks to the Author):

I am fine with revision.

A detailed response to the reviewer is given below. Reviewer comments are in black and our responses are in blue.

Reviewer #1 (Remarks to the Author):

Overall the authors have addressed most of the comments raised by the reviewers. I still have two main comments relative to the assumption made for inter and post seismic modelling, and the way results are presented.

Interseismic loading:

It seems that authors have performed a new model so that GPS data are reasonably fit. However, new figure 4 is not clearly explained. It seems that the authors invert for the interseismic coupling. To what upper and lower locking depths does the inverted model correspond to? This should be mentioned in the text. I also suggest that the authors add the iso-depth contours of the slab in the figure 1.

We appreciate the suggestion to add the iso-depth contours to Fig. 1. Indeed, this clarifies that the locking depth of the inversion model reaches to between ~30 and 40 km depth. This change in presentation thus clarifies that there is no mayor difference between forward and inverse models, as was stated in the text (Line 199).

Revised Fig. 1. Note that the inverse model (colored line) shows full locking downdip to a depth of ~30 km (about where the yellow star is located).

I am still quite puzzled by the difference between the forward models and the inverse model in the predicted horizontal components...

The main difference between both models is the shape of the downdip transition from full locking to free slip. While the forward model imposes a linear transition, the inverse model allows for a more complex transition which fits the inland GPS sites better (Fig. 4). This change in shape results in differences seen mostly in the vertical components (shift in the locus of uplift in lower panels of Fig. 4). In fact, there are no mayor difference among both models in terms of the conclusions of our study, as we stated in the text.

The reviewer acknowledges that this minor differences do not affect the main conclusions of our study.

What if the author propose a forward model with a much shallower locking depth (e.g. 15 or 20 km). This would fit the data much better, and I think it would also correspond to the results of the inversion. I think it is really strange that the authors still chose to show models with locking depths ranging from 30 to 40 km, when the data seem to show something much more shallow. Why do the authors do this? Do they want to avoid showing such a shallow locking? If so why? If the authors have an a priori, it should be explained more clearly...

We performed the test suggested by the reviewer of a forward model with much shallower downdip locking depth. Of course we do not avoid showing such a model and we now include the results of a shallow-locking model to the Supplementary Materials (Fig. S13). The fit is by no means better. In fact, both models predict very similar locking depths; the main difference is that the inverse model predicts a different shape of the downdip transition zone, which results in better fit to the inland data. We believe that the revised version of Fig. 1 including the iso-depth contours of the plate interface will let readers clearly appreciate the locking depth of the inverse model results.

Figure S13. Forward model sensitivity to downdip locking depth. All models include a downdip transition zone of 5 km and a fixed updip depth of 5 km.

The way it is currently presented indirectly suggests to the reader that the locking is full between 5 and 30 km, which is not what is shown by the data.

Figure 1 now clearly shows that the inverse model also predicts full locking to between ~5 and ~30 km depth.

Post-seismic:

The authors discard the possibility of long-term afterslip, but they instead use a very low transient viscosity, which is meant to fit relatively short relaxation time. I am not sure that one can discriminate between both phenomena actually... Long term GPS transient have been observed and modelled as long lasting slow slip on the subduction interface. I do not understand why the authors rule out the possibility of such long-lasting afterslip, since no earthquake of such large magnitude has ever been measured.... I think this is more a principle position from the authors, than any data driven observable...

Indeed, we discarded the possibility of afterslip lasting several decades. We mentioned that afterslip is expected to last for about a decade (based on citing studies from other large-magnitude earthquake in Chile and other subduction zones), and that we cannot precisely assess the timescale as no data showing the duration of afterslip for such a mayor earthquake exists.

Whether the early postseismic deformation can be explained by afterslip only or combination of afterslip and viscoelastic relaxation; or whether the long-term deformation can be explained by combination of afterslip and background viscoelastic relaxation or afterslip with a transient viscoelastic relaxation is still debated (Qiu et al., 2018). Our results provide insight into this debate and agree with Qiu et al. (2018) in suggesting that long-term deformation is predominantly governed by viscoelastic relaxation. Their study used decadal-scale GPS data following the great 2004 Sumatra earthquake (M9.2 – the largest earthquake recorded by GPS) suggesting a very low transient viscosity, in agreement with our results.

We had mentioned this in the text and believe it provides a broad enough context for readers to assess the potential role of afterslip versus mantle relaxation in postseismic deformation.

These remarks however do not change the main conclusions of the paper, which novelty is to be recognized. I therefore recommend publication once my suggestions to clearly show the depth of the inverted boundaries of the interseismic coupling have been taken in consideration.

We appreciate the positive comment.

Reference

Qiu, Q., Moore, J.D.P., Barbot, S., Feng, L., Hill, E.M., 2018. Transient rheology of the Sumatran mantle wedge revealed by a decade of great earthquakes. *Nature Communications* 9, 995.